# Tsunami detection methods for Ocean-Bottom Pressure Gauges

Cesare Angeli[1], Alberto Armigliato[1], Martina Zanetti[1], Filippo Zaniboni[1], Fabrizio Romano[2], Hafize Başak Bayraktar[2], and Stefano Lorito[2]

[1]Alma Mater Studiorum - University of Bologna, Department of Physics and Astronomy "Augusto Righi" (DIFA), Bologna, Italy
[2]Istituto Nazionale di Geofisica e Vulcanologia (INGV), Rome, Italy

**Correspondence:** Cesare Angeli (cesare.angeli2@unibo.it)

**Abstract.** Real-time detection of tsunami waves is a fundamental part of tsunami early warning and alert systems. Several algorithms have been proposed in the literature for that. Three of them and a newly developed one, based on the Fast Iterative Filtering (FIF) technique, are applied here to a large number of records from the DART monitoring network in the Pacific Ocean. The techniques are compared in terms of earthquake and tsunami event-detection capabilities and statistical properties of the detection curves. The classical Mofjeld's algorithm is very efficient in detecting seismic waves and tsunamis, but it does not always characterize the tsunami waveform correctly. Other techniques, based on Empirical Orthogonal Functions and cascade of filters respectively, show better results in wave characterization but they usually have larger residual than Mofjeld's. The FIF-based detection method shows promising results in terms of detection rates of tsunami events, filtering of seismic waves and characterization of wave amplitude and period. The technique is a good candidate for monitoring networks and in data assimilation applications for realtime tsunami forecasts.

. TEXT

## 1 Introduction

Tsunami early warning and alert systems operations are based on the rapid earthquake characterization in terms of magnitude and hypocenter after which alerts are typically given based on a decision matrix or on databases of pre-computed tsunami propagation scenarios; the forecast can be then confirmed, updated, or canceled based on additional earthquake information (e.g., focal mechanism, moment tensor, finite fault models) and sea level measurements (Titov et al., 2005; Duputel et al., 2011; Lomax and Michelini, 2013; Amato et al., 2021). The latter are crucial for the rapid characterization of tsunami waves, monitored from tsunami warning centres by means of coastal tide gauges and/or Ocean-Bottom Pressure Gauges (OBPG) (Rabinovich and Eblé, 2015).

Historically, the first tsunami recording instruments have been the coastal tide gauges, for which self-recording variants have been available since the 1830s (Matthäus, 1972). However, such instruments measure the sea level in close proximity to the coast. For this reason, they are not the primary choice in the the context of tsunami detection, even though they can be used for

early warning purposes wherever no other instruments are located and realtime detection algorithms for tide gauges have been developed (Bressan et al., 2013; Lee et al., 2016). Furthermore, the tsunami evolution at coastal locations is deeply influenced
by its nonlinear interaction with the local bathymetry and topography.

Conversely, these site-effects are negligible in case of measurements of tsunami waves in deep water environments using OBPG. By virtue of being located at the bottom of the ocean, these instruments only detect long waves signals, such as tsunamis and tides, filtering out naturally the most superficial oscillations; moreover, the open ocean tsunami evolution is less affected by complex interactions with coastal morphology and it is mostly a linear phenomenon. Thus, the signals from ocean-bottom
measurements are a superposition of

1. tidal oscillations, dominated by diurnal and semidiurnal periods, which are the main contribution to the energy of the signal;

2. random oscillations in the same frequency range of tides, not accounted for by harmonic analysis;

3. tsunami waves;

4. changes in pressure due to displacement of the ocean bottom;

This last case is evident for gauges subjected to seismic shaking, as for instruments located relatively close to the earthquake source zone, for which seismic and tsunami waves may not be well separated in the recordings, making the extraction of the tsunami wave quite challenging. Thus, techniques able to separate the two contributions are necessary for instruments located near the potential earthquake sources (Williamson and Newman, 2019). For detailed discussions on the nature of deep-ocean
pressure measurements, we refer to the many detailed works in the literature, such as Rabinovich (1997); Goring (2008); Mungov et al. (2013); Rabinovich and Eblé (2015).

OBPG can be classified based on the transmission technology they used, either through cable or through acoustic transmission. Cabled instruments transmit data as soon as they are acquired through cable to research or data centers, where they are analyzed for both early warning applications and later studies. Cabled instruments are commonly used for realtime forecast (Tsushima
et al., 2007) and data assimilation applications (Wang et al., 2019a) and they are used as part of the DONET (Kawaguchi et al., 2008) and S-NET (Mochizuki et al., 2018) networks, deployed around the Pacific coasts of Japan to monitor both local and far field earthquakes and tsunamis.

OBPG with acoustic transmission are the ones used in the Deep-ocean Assessment and Reporting of Tsunamis (DART) network (National Oceanic and Atmospheric Administration, 2005; Titov et al., 2005), composed of a variable in time number
of OBPG operating around the Pacific, northeastern Indian and north Atlantic Oceans, each of which continuously transmits pressure data to a buoy located at the sea surface, which then transmits data to the Tsunami Warning Center. The data transmission frequency between the pressure gauge and the buoy increases in case of an event, which can be triggered by an automatic detection or an external prompt. The recorded pressure changes can then be converted to sea level variation and incorporated into the tsunami forecast with different methods, such as real-time source inversion (Titov et al., 2003; Tang et al.,

2009), data assimilation methods (Maeda et al., 2015; Wang et al., 2017, 2019a, b; Heidarzadeh et al., 2019; Wang et al., 2021), or recently proposed Bayesian approaches (Selva et al., 2021b).

Lastly, we mention that another possible instrument for direct sea level measurement offshore is the GPS buoy (Kato et al., 2000). In this case, a GPS receiver is placed on a stable buoy and data are analyzed at a ground base station using real-time kinematic (RTK) to obtain the relative vertical motion of the buoy. GPS buoys have been used for real-time tsunami inversion as well (Yasuda and Mase, 2013).

The purpose of this work is to test and compare real-time tsunami detection methods from the literature that have been applied to real data acquired by OBPGs. In particular, some techniques are chosen and then applied to past OBPG data *as if* it would happen in real-time. The first technique is the one proposed by Mofjeld (1997). Since every DART station has the algorithm implemented on-board, the technique has a long story of applications and analysis of its properties (Beltrami, 2008, 2011; Chierici et al., 2017). It has to be noted that Fourth Generation DARTs (DART 4G) also include an additional algorithm which allows the automatic separation of seismic shaking and tsunami waves, exploiting the higher sampling rate (https://www.ndbc.noaa.gov/dart/dart.shtml). For that, 1 s sampling rate data are used (Moore, 2024). Since not enough information regarding DART 4G is publicly available, in this study, we do not deal with this algorithm, bur rather test the other algorithms on largely available DART data with sampling times of 15 s. Many of them are currently operational. It has also to be noted that even the on-board sampling rate is higher, the ordinary transmission rate are lower, and the on-board computing capability are generally limited, in both cases to limit the battery consumption. Thus, it is desirable to have a detection algorithms that work for relatively low sampling rates.

The other techniques presented and tested are the detiding through Empirical Orthogonal Functions (Tolkova, 2009, 2010) and Tsunami Detection Algorithm developed by Chierici et al. (2017). Lastly, a new technique, similar to the one developed by Wang et al. (2020), is presented. This new technique is based on the Fast Iterative Filtering (FIF) technique (Cicone, 2020; Cicone and Zhou, 2021) and the IMFogram time-frequency representation (Barbe et al., 2020; Cicone et al., 2024a).

These applications include tests on background signals, i.e. signals where no evident earthquake or tsunami oscillation is present, and on records acquired during the generation and propagation of past events. With these analyses, we are able to characterise each technique in terms of their filtering capabilities for both high and low frequency disturbances. Tests on past events' signals allow us to quantify the detection rates of each technique and to evaluate how they would perform in an early warning setting through simple detection scores. Moreover, we propose simple possible criteria to determine optimal detection thresholds for each detection method, based exclusively on real OBPG data.

The four techniques, which we will refer to as MOF, EOF, TDA and FIF for brevity, are described in their basic mathematical structure in section 2. Applications are then shown in section 3, in order to study how the techniques behave on signals with and without tsunamis.

## 2 Algorithms for real-time tsunami detection

### 2.1 MOF algorithm

DART stations in the NOAA monitoring network are equipped with an automatic tsunami detection algorithm, described by Mofjeld (1997). The algorithm compares the pressure recorded at each instant with a prediction computed from the previously acquired data. If the absolute difference between these two values exceeds a given threshold, this is considered a detection of a sea level anomaly.

The predicted value is found by using Newton's forward polynomial interpolation formula as

$$H_p(t') = \sum_{i=0}^{3} w_i \overline{H}(t - i\mathrm{d}t) \tag{1}$$

where $t'$ is the prediction time set to $t + 5.25\,\mathrm{s}$, where $t$ is the time of most recent measurement used to compute the interpolating polynomial, $\overline{H}$ is the $10-\min$ moving average of pressure data and $\mathrm{d}t = 60\,\mathrm{min}$. For the default parameters, it can be shown that

$$
\begin{aligned}
w_1 &= +1.16818457031250 \\
w_2 &= -0.28197558593750 \\
w_3 &= +0.14689746093750 \\
w_4 &= -0.03310644531250
\end{aligned}
\tag{2}
$$

The technique is particularly suitable for on-board implementation, due to its very simple mathematical formulation, computational efficiency, and low requirements in terms of data needed for the prediction, since little more than the previous $3\,\mathrm{hr}$ of measurements are needed.

We note that the most recent point used for extrapolation is $5\,\min$ before current time. If a tsunami signal has a period longer than that, the averaging operation will not be able to remove it, so the extrapolation will be affected by the presence of the tsunami. The result is that residuals produced by Mofjeld's algorithm deviate in terms of amplitude and period from the tsunami waveform. The problem is addressed by Beltrami (2011), showing that a better agreement between the residual and the tsunami waveform may be obtained by adopting a longer prediction time. However, this results in a much smaller signal-to-noise ratio. The technique has no built-in method to filter out high frequency components, such as random noise and seismic waves.

### 2.2 EOF detiding

The use of Empirical Orthogonal Functions (EOFs) for detiding has been introduced by Tolkova (2009, 2010). The method is based on the application of Principal Component Analysis to a pressure record $\zeta(t)$ as follows:

1. extract from a long time series $N$ segments of $M$ points length;

2. compute the covariance matrix

$$C_{ij} = \sum_{k=1}^{N} \left[\zeta(q_k + i - 1) - a_k\right]\left[\zeta(q_k + j - 1) - a_k\right] \tag{3}$$

where $q_k$ is the index where the $k$-th fragment starts and $a_k$ is the average of the $k$-th fragment;

3. compute the EOFs $e_i$ as the eigenvectors of the matrix $C_{ij} + C_{M+1-i,M+1-j}$.

It is shown by Tolkova (2009) that the first few EOFs are sufficient to reconstruct the tidal component of the sea level signals. Furthermore, Tolkova (2010) shows that these bases have a universality property. In fact, if we compute the EOFs for data obtained in different locations, the residual produced by detiding a signal has the same amplitude whatever basis we use. For this reason, once we have data from a tsunameter in a basin, the technique may be applied to detide any signal from any other instrument within the same basin.

The explantion proposed by Tolkova (2010) hinges on the fact that the periods of the diurnal and semidiurnal tidal components, captured by the first few EOFs are the same in every position in the global Ocean. Since tides are obtained by projection, the differences in amplitude between different basis function sets have no effects on the decomposition. On the other hand, the tidal fine structure, i.e. tidal oscillations of shorter periods such as $6\,\mathrm{h}$ and $8\,\mathrm{h}$, are location specific and are thus removed with this technique. It should be noted also that the *universality* of the main tidal components has been shown empirically, but we lack a rigorous justification. Thus, the property may not be valid for data acquired very far from the sensors used by Tolkova (2010).

To apply the technique to real time tsunami detection, a 1 lunar day long $(24\,\mathrm{h}\,50.4(\mathrm{min}))$ basis is used. At each time step

1. the signal average is subtracted from the data;

2. tides are extracted by projecting the last acquired data onto the EOF basis;

3. a residual is computed by subtracting computed tides from the original signal;

4. the last residual point is compared with a given threshold;

5. once a new measurement is acquired, the computation is repeated on the new 1 lunar day time window that ends at that measurement.

Since tides are obtained by projection, the result of the computations are unaffected by multiplying any of the basis functions by an arbitrary constant.

The technique is computationally efficient, since computing the tides $s$ by projection of a signal $\eta$ can be done by a simple matrix-vector product as

$$s = EE^T\eta \tag{4}$$

where the matrix $E$ has the EOF basis vector $e_i$ as columns. It has also been shown that using 7 elements for the basis minimizes errors for the chosen signal length. In this work, the basis is obtained from the DART 46414 in the period between 06/06/2018 and 08/06/2022, since it presents no discontinuities or missing data. The obtained basis is shown in Fig 1.

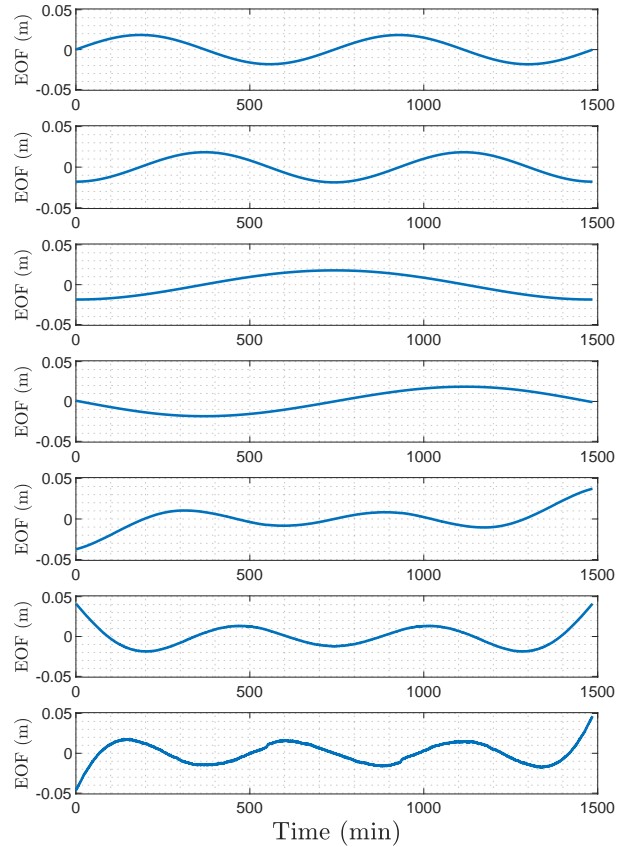

**Figure 1.** Empirical Orthogonal Functions computed for the DART 46414 (in the period 06/06/2018 - 08/06/2022) and used for EOF-based detection. The DART is located South East of Chirikov Island in the Alaskan Gulf. We note that the vertical scale is arbitrary. since the tides are computed by projection on this basis, any multiplicative scaling of any EOF would have no effect on the computed tides.

## 2.3 TDA

Tsunami Detection Algorithm (TDA), introduced by Chierici et al. (2017), has a modular structure, which includes tide prediction and signal filtering. Each time a new pressure measurement is acquired, tides are removed using a harmonic model
(Pawlowicz et al., 2002) with precomputed coefficients. After that, a spike detection algorithm is used to eliminate isolated spikes. Lastly, the residual time history is band-pass filtered using a Finite Impulse Response (FIR) filter. Since the TDA is designed to work in real-time, that is only utilizing previous data, a mirroring boundary condition is applied to the signal before filtering.

The TDA has been specifically developed to be as computationally efficient as possible and the processing at each time step
requires only a few thousands floating point operations. Furthermore, the modularity makes it very easily adaptable to different

operational conditions. On the other hand, requiring precomputed tidal coefficients puts a constraint on the applicability, since a relatively long time series, on the order of a few months, is needed at the position of the instrument. Thus, the technique as described by Chierici et al. (2017) cannot be applied to instruments which have been deployed too recently or have recorded jump discontinuities. Both of these events occur in the case of DART instruments, since they are periodically resurfaced for
maintenance and downloading raw data, then deployed again in a different position. The case of jump discontinuities, due to resurfacing or other reasons, usually requires ad hoc processing, as in the case of very long (e.g., multiannual) trends (Mungov et al., 2013). Techniques to account for these occurrences in real-time need further investigation and are outside the scope of the present work. Whenever such a case is present for a signal in our datasets, TDA is not applied to it. An intermediate situation may occur, where enough data to compute a set of tidal coefficients are available, but not enough to remove tidal oscillations
completely. In these cases, the residual produced by the technique may have amplitudes of several cm that may produce false detections even in absence of any anomaly. Local tidal ranges may also play a role, since areas with much larger tidal ranges are expected to have larger residuals. Given the modularity of TDA, different detiding techniques can be employed in place of the harmonic model presented (Consoli et al., 2014). However, this is outside the scope of the present analysis.

In this work, tidal coefficients are computed using UTide (Codiga, 2011) from at least 2 months of data ending few days
before the time interval of interest in each case. For the harmonic filter, (Chierici et al., 2017) uses a 4000 points FIR pass band filter with a $[2\,\mathrm{min}, 120\,\mathrm{min}]$ or $[4\,\mathrm{min}, 120\,\mathrm{min}]$ period window. Here, we use the second window, since we are mainly interested in applications to tsunami of tectonic origin. Furthermore, filtering using this period band happens to filter out the frequencies which may be contaminated by infragravity waves (Mungov et al., 2013).

### 2.4 FIF-based tsunami detection

The FIF technique (Cicone, 2020) is a data-driven signal analysis technique for decomposing nonlinear and nonstationary signals into simple oscillatory components. The decomposition is additive, so that a signal $s(t)$ can be written as

$$s(t) = \sum_{k=1}^{N} I_k(t) + r(t) \tag{5}$$

where $I_k$ are called Intrinsic Mode Functions (IMFs) and $r(t)$ is the residual. Each IMF satisfies the following properties:

1. the number of zero crossings and the number of relative extrema differ at maximum of one unity;

2. the envelopes of relative maxima and relative minima are symmetric with respect to zero.

A function with such properties can be regarded as a generalization of a Fourier mode $f(t) = A(t)\cos(\theta(t))$, where the amplitude $A(t)$ can vary with time and the phase $\theta(t)$ is allowed to be nonlinear (Huang et al., 1998).

The most common method to decompose a signal into IMFs is the Empirical Mode Decomposition (EMD), introduced by (Huang et al., 1998)). However, the FIF method has some properties that makes it preferable. In particular, it is more
robust to noise (Cicone et al., 2016), it is not prone to mode mixing (Cicone et al., 2024b), and it generates no unwanted oscillations as defined by Cicone et al. (2022). The FIF shares some of these good properties with the Ensemble Empirical

Mode Decomposition (EEMD, Wu and Huang (2009)), but for EEMD this comes with a severe increase in computational cost. On the contrary, FIF can be formulated using FFT (Cicone and Zhou, 2021) making it numerically efficient. To perform a time-frequency analysis of a signal, FIF is complemented by the IMFogram technique (Barbe et al., 2020; Cicone et al., 2024a). From the IMFogram, instantaneous amplitudes and frequencies are computed for each components from the envelope of the absolute value of extrema and the distribution of zero crossings, respectively.

The tsunami detection strategy we propose is as follows:

1. take the last 3 hours of acquired sea level data;

2. remove the long period trend by robust polynomial fit (Street et al., 1988);

3. decompose the residual using the FIF technique;

4. sum the IMFs with frequency content, computed with the IMFogram method, lying within a chosen frequency band;

5. compare the last point of the obtained signal with a chosen amplitude threshold;

6. repeat from step 1 once a new sea level measurement is acquired.

The reason for which the tidal trend is removed through polynomial fit lies in the ability of IMFs to capture components with variable frequency. Just after the arrival of a tsunami wave, as in the example in Fig. 2, using FIF on raw data before detrending may not separate the tsunami wave from tides. For 3 hr long signals, polynomials of degree 3 seem to be the most appropriate. In terms of frequency band, in this work we retain components with periods between 4 min and 180 min, which represent a consevative window for conserving all components related to earthquake generated tsunamis. Nonetheless, we point out that the combination of FIF and IMFogram techniques is quite robust with respect to the choice of both their parameters and the chosen period window. An example of the procedure for one time step is shown in Fig. 2.

The decomposition step in Fig. 2 deserves further comments, in regards to the presence of non-causal oscillations in the components, i.e. non physical oscillations before the tsunami arrival. Firstly, we point out that this effect is not exclusive to FIF (or FIF-like) decompositions and it can be observed also in classical Fourier trigonometric series. One example can be found in Figs. 18 and 19 in the work by Tolkova (2009). Secondly, the tsunami component of the signal is obtained by summing components within the chosen frequency band. In doing so, the additional oscillations cancels out, as shown in the example in Fig. 2. Thus, they have no effect on the obtained residual. Nonetheless, this effect could be avoided using a different set of parameters for the FIF decomposition. Given the robustness of the technique the result of the present work do not depend on this choice. Therefore, the determination of optimal parameters is left for future works.

A similar detection technique based on data-driven signal decomposition was proposed by Wang et al. (2020). The FIF-based detection proposed here differs in two aspects. Firstly, they use the more computational expensive EEMD-based signal decomposition. Despite having two more steps, namely trend removal and frequency computation, than the technique by Wang et al. (2020), the numerical efficiency of FIF makes our algorithm faster overall. Secondly, Wang et al. (2020) a-priori choose which components represent the tsunami, while we choose them based on the frequency content computed at each time step.

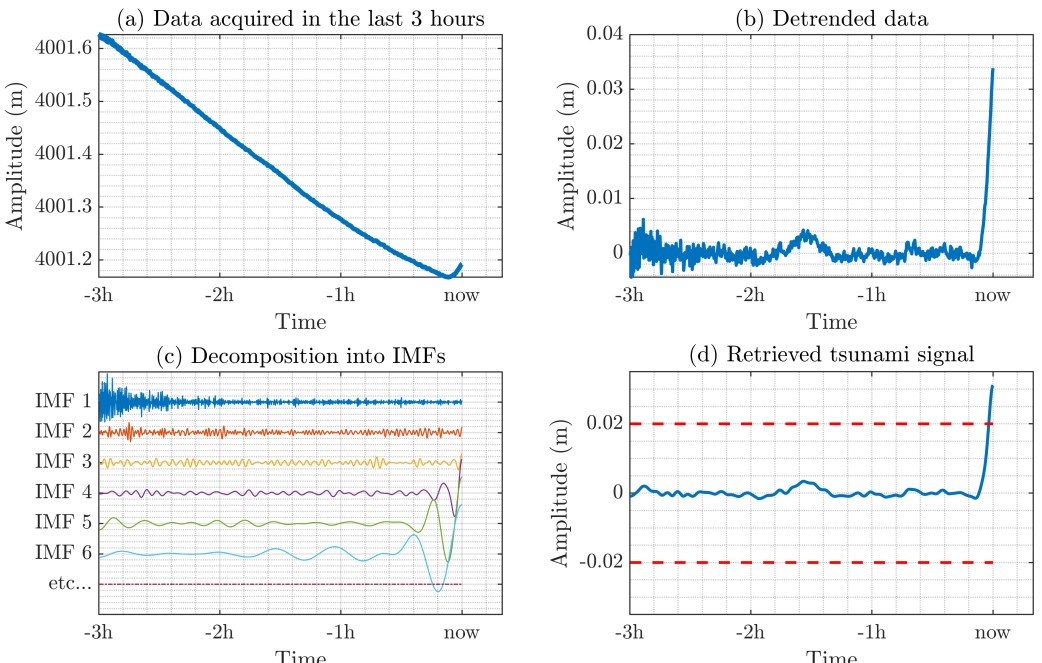

**Figure 2.** Example of FIF-based decomposition: (a) the last three hours of data are taken, (b) detrended through polynomial fitting, (c) decomposed using FIF and (d) the sum of components in the chosen frequency band (periods between 4 min and 180 min) are summed. Data from DART 32413, during the 16/09/2015 Illapel tsunami. The generating earthquake occurred at 22:54:32 UTC, while "now" in the plot refers to 4hr43min after origin time.

## 3 Performance comparison

To test the algorithms described in the previous sections, raw data from OBPG were retrieved from the *Unassessed Ocean Bottom Pressure (highest available resolution)* catalog available on NOAA's website (https://www.ngdc.noaa.gov/thredds/catalog/dart_bpr/rawdata/catalog.html). For background analyses, DART data from New Zealand's network were also used, which are available from GeoNet's website (https://tilde.geonet.org.nz/).

     All the techniques we consider here are amplitude based, that is they process the most recent available portion of data and

a detection is triggered based on the amplitude of the last point of the processed signal. To characterize the properties of each technique, we analyse the time history made of these last points processed at each time step. We will refer to these time series as *detection curves*. The content of the detection curves is a superposition of residuals of the analysis and any component that is not filtered in the processing. However, the specific nature of these contributions differs among the techniques. For example, MOF only removes long term trends, so detection curves contain oscillations from seismic and tsunami waves and random

noise. In the case of techniques with a model-based tide removal algorithm, such as EOF detiding and TDA, there may be contributions to the detection curves due to unmodelled tidal components. The role of random instrumental noise is reduced

wherever high-frequency filtering is used, e.g. for TDA and FIF, but not completely eliminated, since detection curves include only the last point of each analysis and are thus prone to errors from the boundary treatment. FIF detection curves may also be affected by errors in the polynomial fit. In every case, the characteristics that we want from an ideal detection curve is to have an amplitude that increases in correspondence with the passage of a tsunami wave, while remaining below a given threshold anywhere else. Also, it is desirable to have them symmetric around zero. For example, in a detection curve with a negative bias, leading trough waves may be detected even if they have amplitude smaller than the detection threshold, while leading crest waves may not be detected even if larger than the threshold.

Each technique is applied to two different datasets. The first dataset includes time series consisting only of tides and random noise, which we will refer to as *background signals*. Among these we have 5 time series of one month length, where the first characteristics of each technique are shown, and 16 signals from different instruments recorded simultaneously in absence of tsunami events. From these analyses, we are able to characterize the properties of the residual. The second includes day-long signals recorded at DART stations during real tsunami events, to check if the techniques are able to detect tsunamis and if and when a detection is false or triggered by seismic shaking.

We note here that the applications have been carried out on raw pressure data and that all data and plots are expressed in meters of equivalent water, through the equivalence $1\,\mathrm{dbar} = 1\mathrm{m}$. The equivalence is only valid whenever the vertical pressure profile in the water column can be assumed to be hydrostatic. While this is usually the case for earthquake-generated tsunamis, there are cases where it fails, such as the case of coupled air-sea waves (Okal, 2024). While taking this into account is fundamental for the proper characterization of the tsunami source, it does not have an effect on the properties of detection algorithm. However, any integration of the techniques tested here into any alert system must take into account that tsunami waveforms observed in detection curves represent pressure signals and any use in data assimilation or forecast methodology must correctly convert it into sea level time series.

## 3.1 Background signal analyses

For the analysis of the background signal, we started by selecting five time series of one month length, according to the following criteria:

1. no visible seismic or tsunami oscillations;

2. no instrumental spikes, holes or discontinuities;

3. part of a deployment long enough to accurately compute tidal coefficients needed for TDA.

Furthermore, the different time series are taken from instruments installed in different areas around the Pacific Ocean, to avoid biases due to regional features.

Here, we illustrate the analysis by considering the signal chosen from DART 46414 (Fig. 3) whose detection curves for each technique are shown in Fig. 4. All detection curves show some residual oscillations, though of different amplitude and spectral content, as shown by the spectra in Fig. 5. The EOF detection curve has peaks for periods around $24\,\mathrm{hr}$, that is the residual

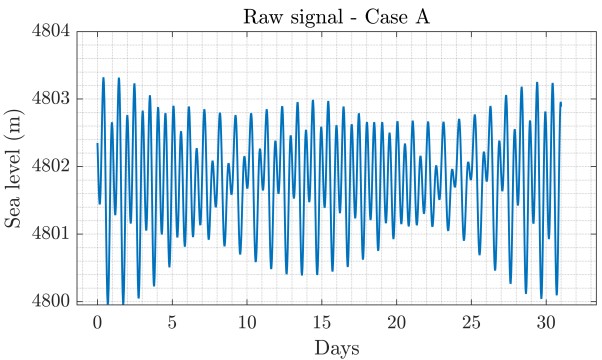

**Figure 3.** Raw data from DART 46414 for the month of August 2019.

**Table 1.** Maximum peak-to-peak amplitude, indicated as $A$, of raw data and maximum variability range of detection curves for each technique in each background case. All quantities are expressed in $cm$.

| Case | DART | Period | $A$ | MOF | EOF | TDA | FIF |
|------|------|--------|-----|-----|-----|-----|-----|
| A | 46414 | 01/08/19 - 01/09/19 | 337.25 | $[-0.59, 0.51]$ | $[-1.39, 1.45]$ | $[-2.16, 1.75]$ | $[-0.92, 1.13]$ |
| B | 52402 | 01/06/16 - 01/07/16 | 82.04 | $[-0.40, 0.42]$ | $[-1.67, 1.85]$ | $[-0.63, 1.25]$ | $[-0.63, 0.71]$ |
| C | 32413 | 01/01/19 - 01/02/19 | 129.62 | $[-0.54, 0.58]$ | $[-1.72, 1.84]$ | $[-1.30, 0.75]$ | $[-0.74, 0.74]$ |
| D | 51407 | 15/04/22 - 15/04/22 | 88.70 | $[-0.51, 0.54]$ | $[-1.94, 1.93]$ | $[-1.27, 0.82]$ | $[-0.94, 0.97]$ |
| E | 21413 | 01/06/21 - 01/07/21 | 86.06 | $[-0.44, 0.43]$ | $[-1.71, 1.62]$ | $[-0.69, 1.46]$ | $[-0.64, 0.63]$ |

from the main diurnal component, and $8\,hr$, which may be related to the fine structure of tidal oscillation, which is not captured by the algorithm (Tolkova, 2010). TDA has the main periods around $12\,hr$ and $24\,hr$, showing that the main contribution to the residual is given by the difference between predicted and observed tides. MOF and FIF detection curves also have a spectral peak around a period of $12\,hr$, but with a lower amplitude. Also, contrary to EOF, they have a mostly flat spectrum far from the semidiurnal frequency band.

It is also interesting to look at the amplitude distribution of the prediction around zero. From the histograms in Fig. 6, we can notice that the MOF technique has the narrowest distribution, since it is able to remove the long-term trends entirely, with the only contribution to the detection curve being the random noise, as pointed out before. FIF has a very peaked distribution, indicating that the detection points checked at each time step do not deviate much from zero. On the other hand, EOF and TDA have a wider distribution, which shows that a larger number of values are further away from zero. For TDA, it can also be noticed that the histogram is not centered around zero, that is the points in the detection curve are distributed asymmetrically, due to the fact that the predicted tides are above the raw data.

The conclusions made for the case in Fig. 3 can be generalized to the analysis of the other 4 background signals. The variability of the detection curves for all 5 cases can be measured by their maximum range of variability and standard deviation, reported in Tab. 1 and Tab. 2, respectively. In each case, MOF produces the narrowest distributions, both in terms of maximum

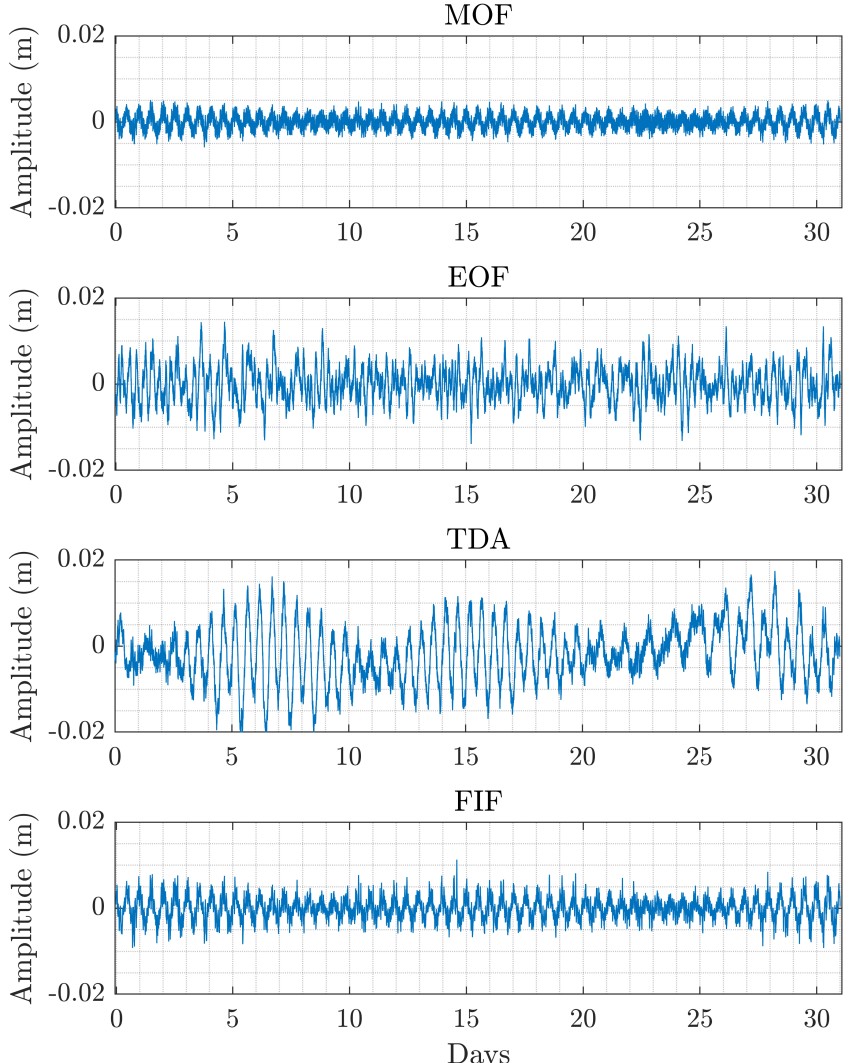

**Figure 4.** Detection curves for each detection technique for DART 46414, August 2019 (Fig. 3).

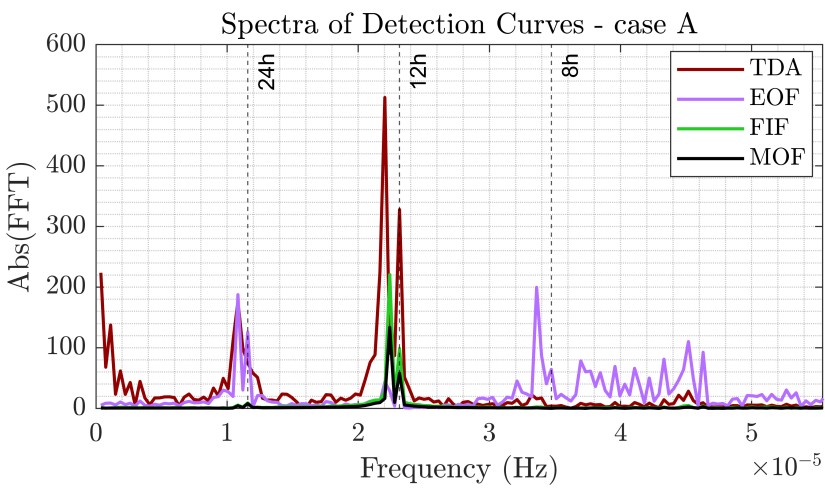

**Figure 5.** Spectra, computed as absolute values of the Fast Fourier Transform (FFT), for each detection curve in Fig. 4, relative to data from DART 46414, August 2019 (Fig. 3).

**Table 2.** Standard deviation expressed in cm for each technique in each background case.

| Case | DART | Period | $A$ | MOF | EOF | TDA | FIF |
|------|------|--------|-----|-----|-----|-----|-----|
| A | 46414 | 01/08/19 - 01/09/19 | 337.25 | 0.15 | 0.41 | 0.64 | 0.24 |
| B | 52402 | 01/06/16 - 01/07/16 | 82.04 | 0.10 | 0.56 | 0.27 | 0.15 |
| C | 32413 | 01/01/19 - 01/02/19 | 129.62 | 0.13 | 0.61 | 0.29 | 0.17 |
| D | 51407 | 15/04/22 - 15/04/22 | 88.70 | 0.13 | 0.63 | 0.30 | 0.20 |
| E | 21413 | 01/06/21 - 01/07/21 | 86.06 | 0.10 | 0.57 | 0.35 | 0.14 |

variability, which is around 0.5 cm from zero, and in terms of standard deviation. FIF shows a larger variability for both metrics compared to MOF, but lower than the other techniques, remaining within 1.2 cm from zero at each time. EOF is the technique with the largest standard deviations from the origin on average, followed by TDA, then FIF and MOF, with the exception of case A, where TDA has larger standard deviation than EOF. This may be caused by the large tidal range in case A (see Tab. 1), which results in less accurate tide prediction.

The asymmetric amplitude distribution of TDA is observed in all cases. Cases A, C, and D are negatively skewed, while cases B and E are positively skewed. On the contrary, the other techniques are approximately symmetrical in every case. For these 5 time series, the threshold of 3 cm, commoly used in DART OBPGs (Mofjeld, 1997; Rabinovich and Eblé, 2015), produces no false detection and seems to be a highly conservative choice. In fact, a 2 cm threshold would result in false detections only for TDA in case A. For MOF and FIF, the threshold may be lowered to 1 cm and 1.5 cm respectively. Plots for cases B, C, D and E analogous to Fig. 3 to 6 are reported in the Supplementary Materials. Furthermore, the analysis of these five cases has been used to have a first order evaluation of the sensitivity of TDA to the amount of data used to compute tidal coefficients. We

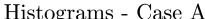

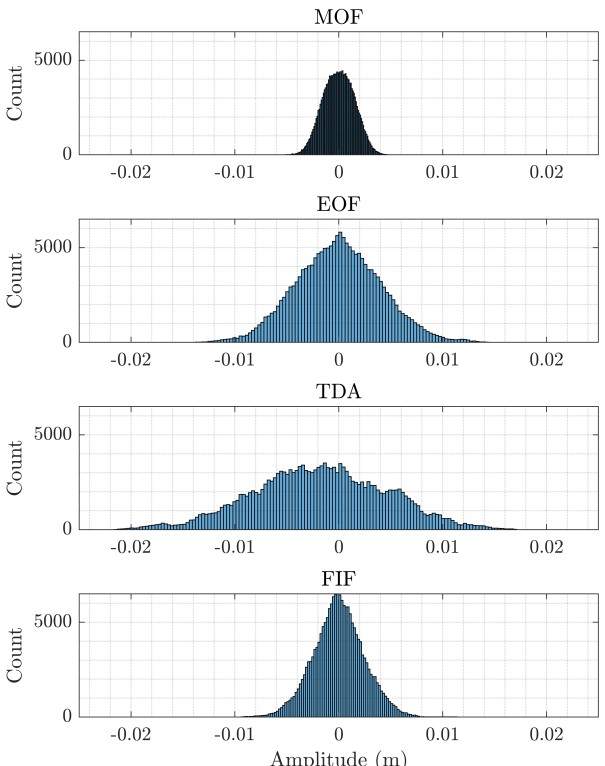

**Figure 6.** Histograms of each detection curve in Fig. 4, relative to data from DART 46414, August 2019 (Fig. 3).

determined that an amount of data between 7 to 9 months results in the narrowest detection curves for TDA. Details about this analysis are reported in the Supplementary Materials.

We should point out that the presented background analysis takes into account geographical variability only weakly, since the 5 signals are from different points in time. Furthermore, multiple tsunamis occurred and were detected during those time windows. This in principle may lead to misinterpretations of the results, weak anomalous oscillations could be attributed to those events. To verify the properties of the detection algorithm accounting for these factors, we set up another test. In this case, we selected data following these criteria:

– signal were acquired simultaneously by different DART stations;

– the considered DART stations were located in various locations to have maximum geographical coverage;

– the raw data preceding the signal at each station should be long enough to accurately compute tidal coefficients for TDA;

– the considered time window should include no tsunami in the regions where the DART stations are located.

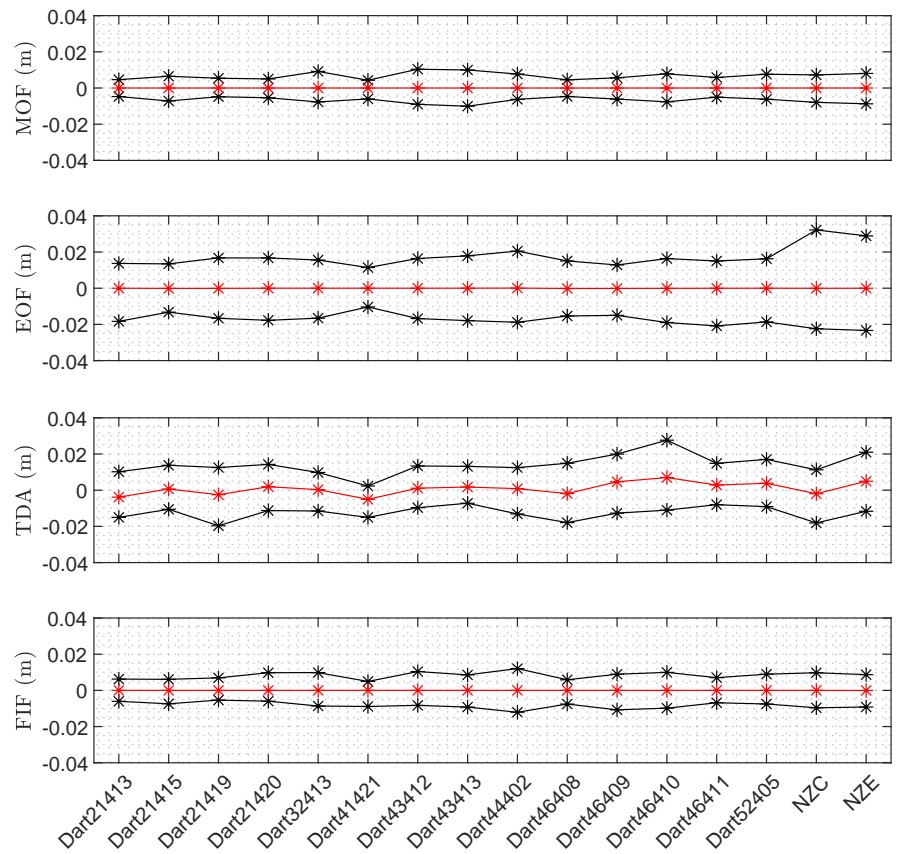

**Figure 7.** Signal average (red) and maximum and minimum amplitude (black) recorded in each detection curve for all signals in the simultaneous background test. The detection curves have been computed for each of the 16 DART stations reported on the $x$ axis over the period 8/04/2021 - 27/04/2021. Note that MOF and FIF have a narrower and more symmetric distribution than EOF and TDA. TDA shows significant asymmetries.

Accordingly, we chose the period between 8[th] April 2021 to 27[th] April 2021. We use data between the 1/09/2020 and 1/04/2021 to compute tidal coefficients. In this period, we found 19 DARTs that met the data amount requirements, 3 of which were excluded due to the presence of isolated spikes in the considered time frame. During this period, only one tsunami is reported by NOAA National Centers for Environmental Information's global tsunami catalog, related to the eruptive activity of La Soufrière volcano on the island of Saint Vincent. The catalog reports a maximum runup of $0.1\,\mathrm{m}$ on the island. Since only two of the 16 stations are in the Atlantic Ocean and the closest is located at $\sim 1000\,\mathrm{km}$, we assume that the signals are unaffected by the tsunami waves. In Fig. 7, we show the signal average and the absolute extrema of each detection curve for each technique. For the most part, the results reproduce what we observed in the previous background examples. MOF and FIF have narrower


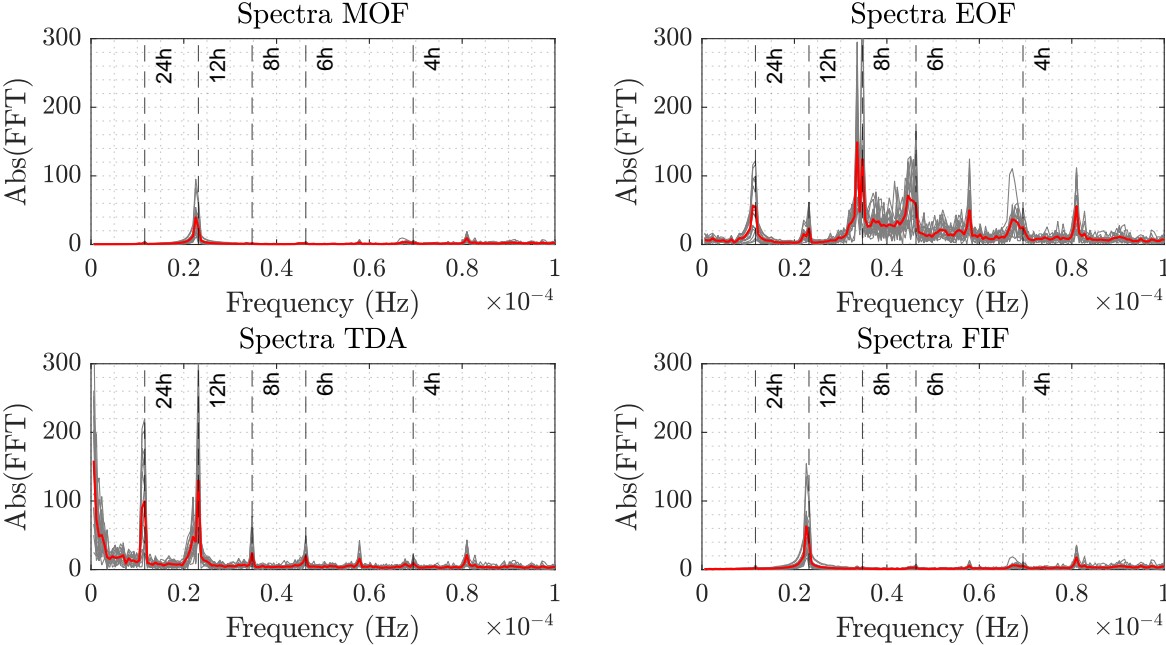

**Figure 8.** Absolute value of the FFT of each detection curve (gray) and average absolute value of the FFT per technique (red). The detection curves have been computed for each of the 16 DART stations reported on the $x$ axis of Fig. 7 over the period 8/04/2021 - 27/04/2021. Note that MOF and FIF show the weakest spectral peak, around the period of semidiurnal tides. EOF and TDA have more pronounced spectral peaks in correpondance of various periods. Signal-long trends in TDA are evident from the larger values in the zero frequency limit.

and more symmetric distributions around zero than EOF and TDA, with no curve that ever reaches an absolute amplitude of 1.3 cm. On the other hand, TDA's signal averages tend to be larger in absolute value, meaning that the amplitude distribution of the amplitudes is skewed away from zero. The result for EOF in the case of New Zealand's DARTs are noteworthy, since they have the largest amplitude residual out of all the background tests: 3.2 cm and 2.9 cm maxima for stations NZC and NZE, respectively. A possible reason might be that these stations are significantly further away than most of the others from the

position of the DART used to compute the EOF basis, located in the Alaskan-Aleutian arc, and the New Zealand area was not covered in the empirical tests by Tolkova (2010), since data were not available at the time. Although one reason for the difference could be found in the distance between these stations and the ones used for the basis computation, we also point out that such large oscillations are not observed in DARTs 41421 and 44402, located in the Atlantic ocean, where tidal regimes can be quite different.

From a spectral point of view, each techniques shows consistent results across different instruments. In Fig. 8, we plot the absolute value of the Fourier transform. For each technique, we also plot the average spectra across the different DARTs. All four techniques show consistent frequency peaks, with MOF and FIF showing peaks around the 12 h periods corresponding to semidiurnal tide period range. EOF has strong peaks for period of 24 h, 8 h, 6 h and 4.8 h, corresponding to tidal oscillations

not well modelled by the EOF basis. On the other hand, the peak at 12 h is weaker than for the other techniques. At last, TDA
has strong peaks at diurnal and semidiurnal periods, meaning that the harmonic fit does not capture the entirety of the main tidal
oscillations. Furthermore, it is the only technique with significant amplitude at the zero frequency limit, showing the presence
of signal long trends that are not eliminated by neither the tide forecast, nor the bandpass filter.

## 3.2  Computational cost

Since the main application of a tsunami detection algorithm is in an early warning context, we require the computational time
to be low (Beltrami, 2008, 2011). The techniques presented in this work are all applied every time the instrument acquires a
new measurement, thus the computational costs for the application to one step must be lower than the acquisition sampling
time, which for the DART stations we considered in this study is 15 s. Another common characteristics of these techniques is
that each should take constant time per step, since it performs the same operations in each step. Thus, we can characterize the
computational cost of each detection algorithm by computating the time per step.
To this aim, we computed the time each technique takes to compute the first 5000 data points of the detection curves for the
background signal A (Fig. 3), reported in tab.3. In all cases, we took into account only computations that are needed in realtime,
while ignoring every computation that may be performed offline, such as the computation of the basis functions for EOF, or the
tidal coefficients for TDA. Computations have been carried out on an Intel Core i9−11900 CPU, 2.50 GHz, 32.0 GB RAM,
Windows 11 Pro and Matlab R2024a.

**Table 3.** Total computational time and time per step in s required for each technique to compute the first 5000 points of the detection curves
in fig.4

| Technique | Time | Time per step |
|---|---|---|
| MOF | 0.001 | $2 \times 10^{-7}$ |
| EOF | 46.89 | 0.094 |
| TDA | 0.005 | $10^{-6}$ |
| FIF | 325.27 | 0.0651 |

For all techniques the computational time per step is much smaller than the sampling time. Thus, they may all be used for
real-time tsunami detection as currently implemented. However, further considerations are needed for real world applicability.
We note that MOF and TDA require orders of magnitude less time than the other techniques. MOF includes just few tens of
floating point operations, for which modern hardware are highly optimized. The same is true for TDA, where the computationally
heaviest operation, i.e. the fitting procedure to determine tidal coefficients, is carried out offline. On the other hand, EOF
detiding and the FIF-based detection technique need 3 to 4 orders of magnitude more computing time. While they are still
fast enough to be used for realtime tsunami detection, their applicability to instruments with an autonomous power supply,
e.g. DART stations, can be limited by the comparatively larger computational cost.

In regards to the FIF-based detection method, we should also point out that its current implementation described in section
2.4 could be optimized. In fact, we note that a complete decomposition of the signal is computed at each time step. However,

the computation could be stopped once the first component with frequency below the chosen frequency band is extracted. Moreover, the parameters of both the FIF decomposition and the IMFogram algorithm have not been optimized for computational costs. At last, the detrending step, carried out with a polynomial fit, might be carried out through more numerically efficient techniques, such as smoothing FIR filters (Schafer, 2011).

### 3.3    Detection testing on past events

To test how the various algorithms compare in the detection of real events, a dataset based on the catalog by Davies (2019) has been built. The catalog includes 18 events that occurred around the Pacific Ocean between 2006 and 2016, generated by earthquakes of magnitude between 7.8 and 9.1. For each event, we extracted 24 hr long signals starting from the earthquake origin time from every DART stations active at origin time whose data are available on NOAA's website. It may happen that data may not be retrieved from some instruments. In these cases, only data transmitted in real-time by the instrument are

available. For the DARTs concerned here, raw data's sampling time is 15 s, while it varies for transmitted data between 15 min for normal conditions, 15 s and 1 min for the 4 hr after a detection is triggered (Rabinovich and Eblé, 2015). For this reason, the cases where only transmitted data are available have been excluded. Finally, we removed instrumental spikes and resampled by linear interpolation. The dataset obtained contains 437 signals of various nature: some of them consists of only background, while others also contain seismic and/or tsunami waves.

A first comparison between the different techniques can be made by computing the average and standard deviation of each detection curve. The comparison among absolute values of the averages (Fig. 9) give results similar to the analysis of backgrounds in the previous section: MOF usually produces the smallest residuals followed by FIF, EOF and at last TDA.

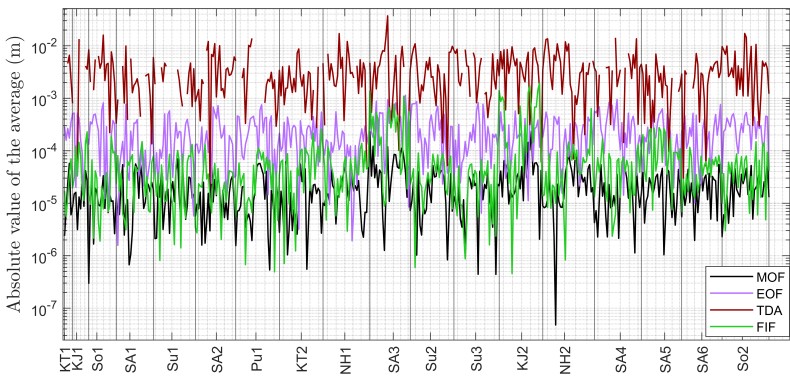

**Figure 9.** Absolute average value of detection curves for each of the 437 signals in the dataset built by including signals for each event in the catalog by Davies (2019) from every active DART stations whose raw data are available through NOAA's *Unassessed Ocean Bottom Pressure (highest available resolution)* catalog. Detection curves are obtained by applying each of the four techniques to every signal in the dataset. The event to which the signals corresponds to are reported along the $x$ axis in chronological order using the nomenclature by Davies (2019). Missing values in the case of TDA corresponds to signals for which there are not enough data for the tidal fit to converge.

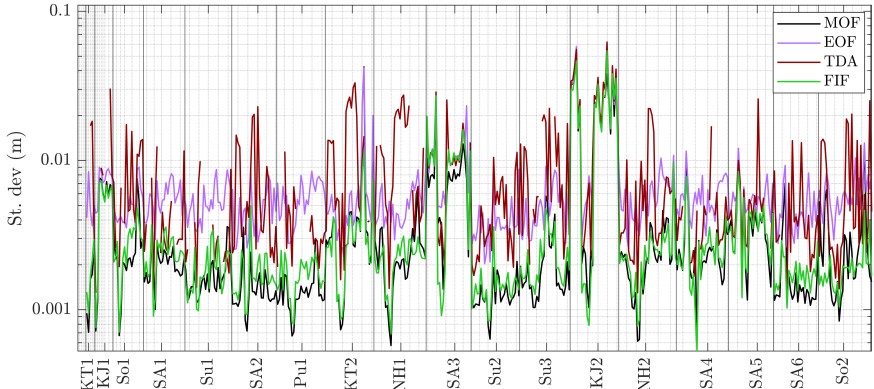

**Figure 10.** Standard deviation value of detection curves for each of the 437 signals in the dataset built by including signals for each event in the catalog by Davies (2019) from every active DART stations whose raw data are available through NOAA's *Unassessed Ocean Bottom Pressure (highest available resolution)* catalog. Detection curves are obtained by applying each of the four techniques to every signal in the dataset. The event to which the signals corresponds to are reported along the $x$ axis in chronological order using the nomenclature by Davies (2019). Missing values in the case of TDA corresponds to signals for which there are not enough data for the tidal fit to converge.

TDA has a worse perfomance than before due to the variable availability of preceding data, as explained in section 2.3. Thus, TDA was not applicable to 55 signals whose deployment was too recent to compute tidal coefficients and it shows large
residuals if the coefficients were computed from relatively short series. Standard deviations are correlated with the total area between the curve and horizontal axis, i.e. we expect the standard deviation to be proportional to amplitude of oscillations in a signal. Thus, the standard deviations of detection curves with low amplitude seismic and tsunami components behave similarly to curve averages, as shown in Fig. 10. On the contrary, for signals with high amplitude seismic and/or tsunami components, the various techniques tend to provide similar results. This is evident for the case of the two Japan-Kurils and the Maule events
(KJ1, KJ2 and SA3 respectively in the plot, see Davies (2019)).

To compare the four techniques, we now try to discriminate false detections and detections triggered by seismic shaking or the tsunami wave. We define for a given detection threshold $T$

- $N$: total number of signals in the dataset;

- $n_F$: number of signals with at least one false detection;

- $n_E$: number of signals with no false detection and at least one earthquake detection;

- $n_T$: number of signals with no false detection and at least one tsunami detection;

- Detection score 1: $\theta_1 = \dfrac{n_T - n_F}{N}$;

- Detection score 2: $\theta_2 = \dfrac{n_T - n_E - n_F}{N}$.

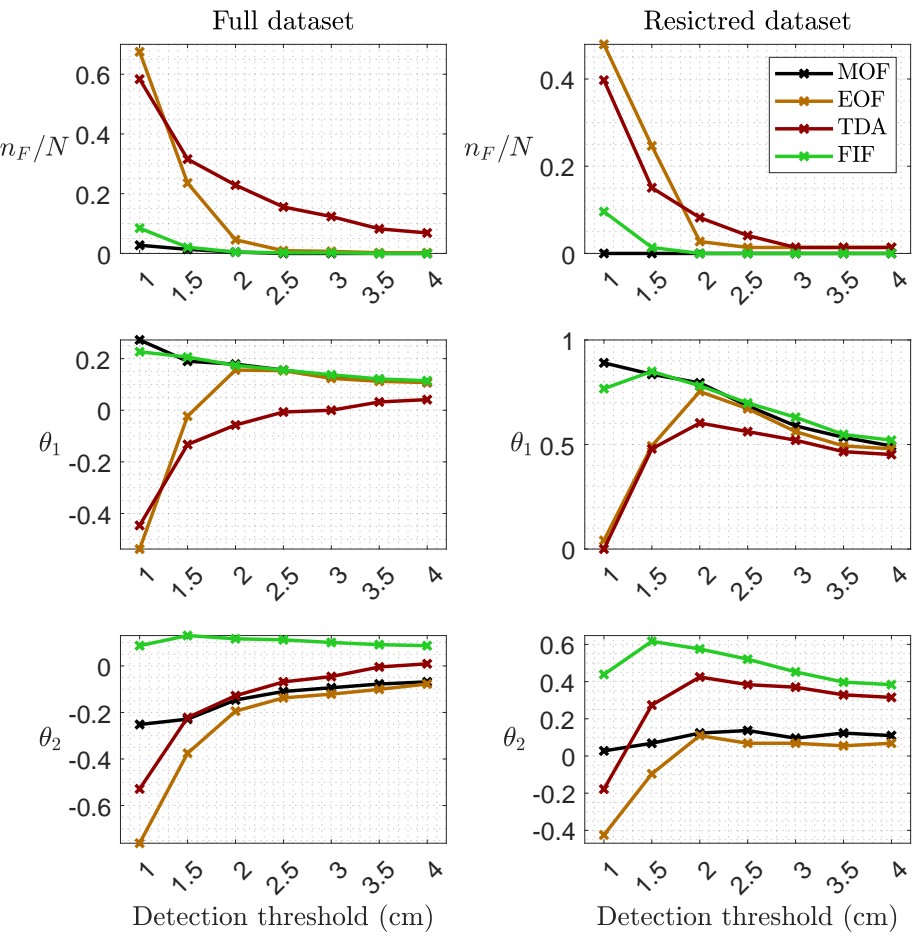

**Figure 11.** Normalised false detections $n_F/N$ and detections scores $\theta_1$, $\theta_2$ for each detection technique for the full dataset (left column) and the restricted dataset of time series available in Davies (2019).

These parameters have been computed for each threshold from $T = 1.0\,\mathrm{cm}$ to $T = 4.0\,\mathrm{cm}$ with a step of $0.5\,\mathrm{cm}$. We note that the process of attributing a detection to the seismic or tsunami wave trains has been carried out by visual inspection. To avoid possible biases, two strategies are employed. First, the attribution has been as conservative as possible, that is any doubtful detection is considered a false detection. Second, we compared detection curves with post-processed waveforms made available by Davies (2019). To obtain these waveforms, Davies (2019) uses a LOESS smoother to find tides, while seismic waves are removed by truncating the time series. The signals in the dataset for which these are available are 73. Furthermore, the analysis has been applied separately to the set of signals for which we have post-processed waveforms, which we refer to as *restricted dataset*, and to the full dataset.

The differences among techniques may be highlighted by comparing the number of false detections and the detection scores, reported in Fig. 11 as functions of the detection threshold. The number of false detections decreases monotonically with the

detection threshold, as we expect. MOF and FIF both have zero false detections above a given threshold, namely $2.5\,\mathrm{cm}$ for the full dataset and $2.0\,\mathrm{cm}$ for the restricted one. On the contrary, EOF and TDA have false detections for each threshold among the ones considered. It is interesting to notice that both reach an asymptote in the restricted dataset for threshold bigger or equal to $3.0\,\mathrm{cm}$. This is also the case in the full dataset for EOF, but not for TDA. The reason is that the full dataset contains a higher percentage of signals with larger tidal residual, which have an amplitude of several centimeters.

The behaviour is different in the case of the $\theta_1$ and $\theta_2$ scores. For $\theta_1$, which can be interpreted as a measure of successful tsunami detections relative to the number of signals with false detections, we observe that MOF gets always better with lower thresholds. On the other hand, for EOF $\theta_1$ has a maximum for a threshold of $2.0\,\mathrm{cm}$. This can be interpreted as the optimal threshold for EOF if $\theta_1$ is assumed to be a good performance metric. TDA and FIF show a slightly different behaviour between the two datasets. FIF has an optimal threshold $T = 1.5\,\mathrm{cm}$ for the restricted dataset. TDA has the same optimum $T = 2.0\,\mathrm{cm}$ as EOF for the restricted dataset, while in the full dataset the presence of signals with large residual dominates as in the previous case.

$\theta_2$ is similar to $\theta_1$, with the added goal to minimise the number of earthquake detections. In the restricted dataset, MOF and EOF perform worse than TDA and FIF, since the latter two filter out the high frequency content. Exactly as it happens for $\theta_1$, EOF and TDA reach optimal score values at $T = 2.0\,\mathrm{cm}$ and FIF does at $T = 1.5\,\mathrm{cm}$. However, in the full dataset FIF is the only technique with an optimal threshold, again equal to $1.5\,\mathrm{cm}$. For the other techniques, $\theta_2$ increases monotonically with the detection threshold. While for TDA the reason is the same as before, MOF's and EOF's performance is dominated by the larger amount of recorded seismic waves.

### 3.4 Waveform characterisation in FIF-based detection

According to Beltrami (2008), one of the desirable properties of a tsunami detection algorithm is the correct characterisation of the wave in terms of amplitude and period. While these properties have already been investigated for MOF (Beltrami, 2011), EOF (Tolkova, 2009, 2010) and TDA (Chierici et al., 2017), they have yet to be established in case of FIF. In particular, we are interested in determining

1. the behaviour on signals where no earthquake or tsunami is present;

2. how seismic waves are filtered and the separation between Rayleigh waves and tsunami waves in the near field;

3. if we can determine the correct tsunami waveform from the detection algorithm.

Regarding point 1, detection curves generally behave as expected from the analysis of the background signal (see section 3.1), with amplitudes mostly within $1.0\,\mathrm{cm}$ and no strong residual oscillation.

For point 2, we already pointed out in section 3.3 that FIF filtering capabilities are well illustrated by the variation of $\theta_2$ as a function of the detection threshold. However, even when the earthquake is detected, FIF allows to better separate seismic and tsunami waves. This is exemplified in Fig. 12, showing the application of the techniques to DART 51425 record during the 29/09/2009 Samoa earthquake and tsunami. In this case, all four techniques would trigger a detection at the passage of seismic

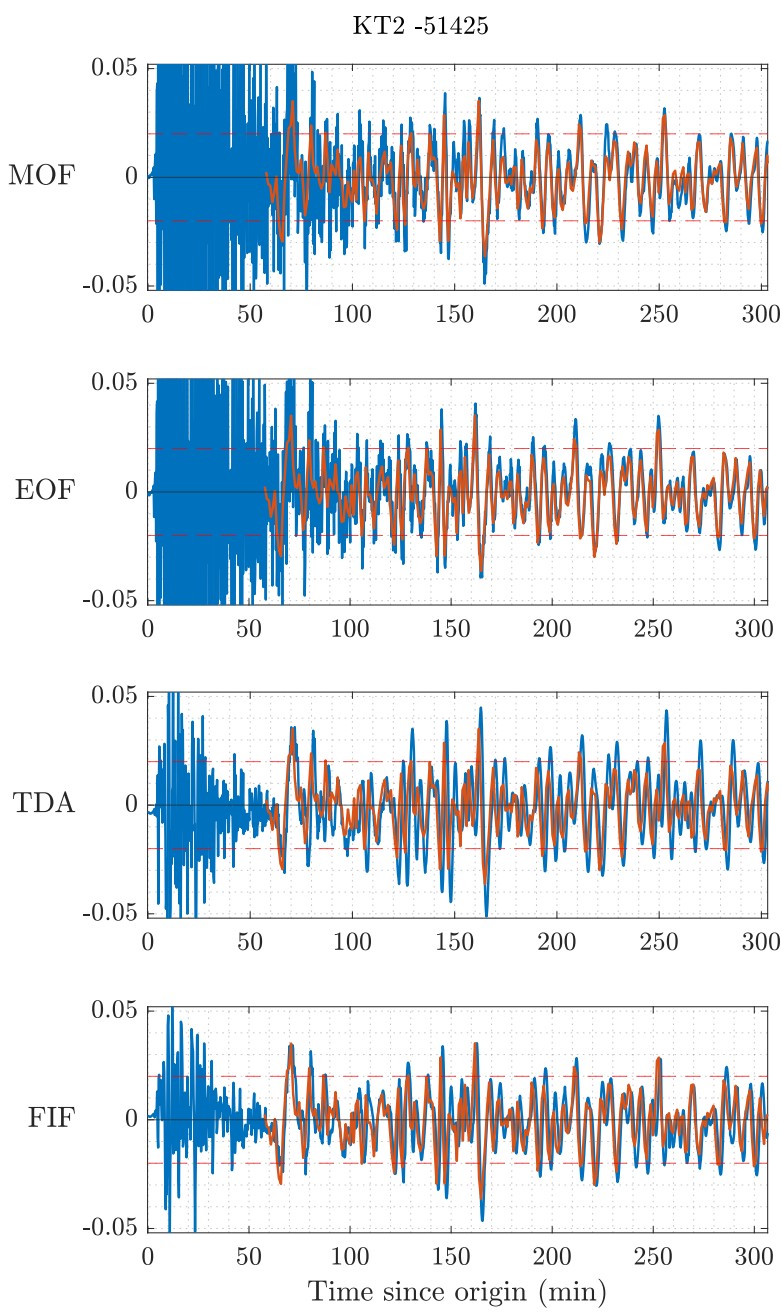

**Figure 12.** Comparison of detection curves (in blue) nand post-processed tsunami waveform (orange) for the 29/09/2009 Samoa tsunami as recorded by DART 51425. Dashed, horizontal, red lines are located at $\pm 2\,\text{cm}$.

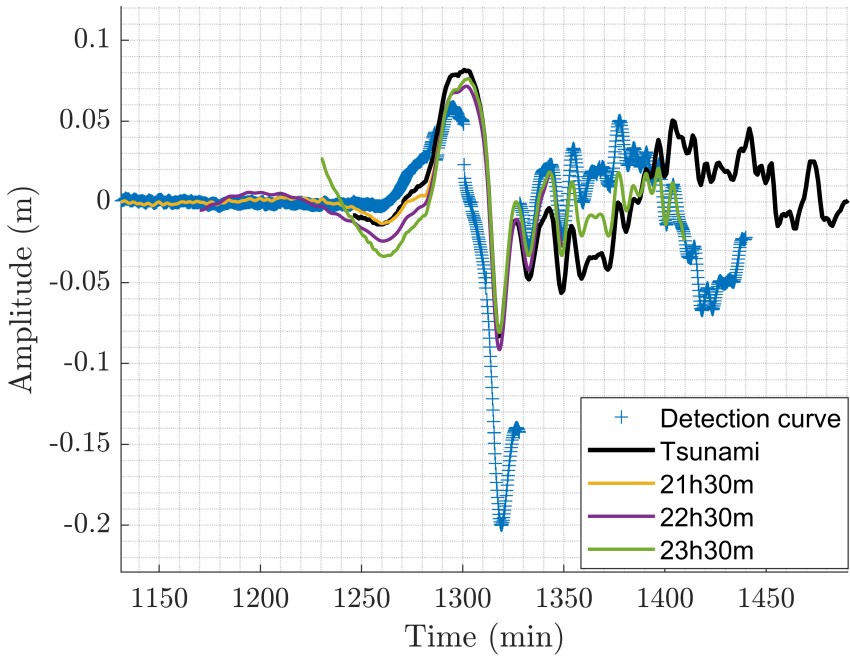

**Figure 13.** Example of the tsunami components (yellow, purple, and green curves) extracted during monitoring compared with the post-processed waveform (black) and the detection curve (blue crosses) obtained with the FIF-based technique. Time is measured from the earthquake origin time. Data from the record at DART 21413 for the 2010 Maule Tsunami.

waves, but the corresponding amplitude varies a lot between MOF and EOF ($\sim$85 cm, not shown in the figure) and TDA and FIF ($\sim$10cm). Furthermore, while for the first two the seismic wave train overlaps with the tsunami wave, for the last two there is a clear separation, allowing for a better estimation of the tsunami amplitude, which represents an important observable that is part of a tsunami alert statement.

In the analysis, we found a limited number of signals where FIF detection curves present jump discontinuities during the tsunami passage, in cases where the signal is very steep. Since the occurrence of these discontinuities only happen during a tsunami, they do not hinder in any way the detection capabilities of the technique. However, such cases may be problematic in data assimilation applications, where the full waveform is needed. In these cases, we can use the tsunami component produced by the decomposition (step 4 in the procedure described in section 2.4) at the time of assimilation.

As it is shown in Fig. 13, the tsunami components extracted during monitoring at different times approximate the tsunami waveform much better than the detection curve by itself. However, the use of the full component extracted through FIF decomposition may be a heavy operation to perform in real time, since it would require transmission of a 3 hr long signal, that is a 720 element long vector, instead of a single number as needed for the detection curves. In instruments where power management is critical, such as DART stations, this operation should be performed rarely, e.g. once at a fixed time after

detection, if precise data are needed, as is the case in data assimilation contexts (Wang et al., 2019b). At last, we also note that such large discontinuities in the detection curve are present in very few detection curves and that the example in Fig. 13 is the most pathological.

## 4   Conclusions

Four tsunami real-time detection algorithms, one of which presented in this work for the first time, have been analyzed and
compared. In particular, they have been tested against a large amount of real OBPG data from NOAA's and GeoNet's DART networks, both in the presence and in the absence of oscillations related to the earthquake and the tsunami. Firstly, we have determined the main properties of the techniques by analysing their application to background signals. These background tests, which include 5 signal of one month duration and signals of 20 days duration acquired simultaneously on 16 different stations, show consistent results in terms of amplitude and spectral content. After that, a dataset of tsunami signals from past events
has been analysed and detection rates of each technique have been quantified through simple detection scores, with the goal of determining optimal detection thresholds.

    For detection applications only, Mofjeld's algorithm remains the best performing technique, both in terms of detection metrics and computational speed. However, the algorithm is not the most suitable to correctly characterize the tsunami waves, nor to filter out high frequency components (e.g. the seismic Rayleigh waves). The EOF and TDA techniques present variable
behaviour. EOF is not able to reduce the tidal residual below $\sim 2\,\mathrm{cm}$, leading to incorrect characterization of low amplitude tsunami signals. TDA has a strong dependence on the precision of pre-computed tidal coefficients, resulting in a large number of detection curves with amplitude of several centimeters, too large for a precise detection of offshore travelling tsunamis. Investigating a combination of TDA with a different detiding technique may be the subject of future work.

    The newly developed FIF-based detection method possibly shows the best compromise between detection and real-time
characterisation. Optimal detection thresholds for the technique have been determined to be

1. $T = 2\,\mathrm{cm}$ for the goal of minimizing false detections;

2. $T = 1.5\,\mathrm{cm}$ for maximizing tsunami detection w.r.t. earthquake and false detections, based on two simple detection scores.

Furthermore, it is shown that the entire tsunami component over the three hour period reproduces accurately the tsunami waveform, allowing the characterization of wave amplitude and period even in the rare cases where the detection curves fail to
do so.

    Future work is planned for the application of the technique to the 4G DART stations and non-OBPG data (e.g. coastal tide-gauges), and to tsunamis of nonseismic origin, for example for OBPGs which are planned at Stromboli to monitor volcano-induced tsunamis (Selva et al., 2021a). On the other hand, the technique is already fast enough to be applied in real-time, but an on-board implementation will require greater optimization to limit power consumption, especially in the case where the entire
tsunami components has to be transmitted. Future work is then also planned for the numerical optimization of the technique by exploiting the recent installation of SMART cables (Howe et al., 2019) and also in view of the recent installations in the Ionian

Sea of a dedicated instrumented cable to detect earthquakes and tsunamis (Marinaro et al., 2024), and of further DART-like OBPGs by CAT-INGV (Amato et al., 2021).

. **Code and Data availability.** All data used in the work are available in the *Unassessed Ocean Bottom Pressure (highest available resolution)* catalog available on NOAA's website (https://www.ngdc.noaa.gov/thredds/catalog/dart_bpr/rawdata/catalog.html) or through GeoNet's Tilde API (https://tilde.geonet.org.nz/). The computation of tidal coefficients has been carried out using UTide (Codiga, 2011), available at https://www.po.gso.uri.edu/~codiga/utide/utide.htm. For the FIF technique (Cicone, 2020) and the IMFogram algorithm (Barbe et al., 2020; Cicone et al., 2024a), we used the codes developed by the original developers of the techniques, available at https://github.com/
Acicone/. Everything else, such as the FIR filter coefficients and the Empirical Orthogonal Functions, has been computed through native MATLAB functions and scripts are available as Supplementary Materials.

. **Authors contributions.** Cesare Angeli: conceptualization, data curation, software, methodology,writing - original draft. Alberto Armigliato: supervision, methodology, writing - review and editing. Filippo Zaniboni: writing - review and editing. Martina Zanetti: software. Fabrizio Romano: methodology, writing - review and editing. Hafize Başak Byraktar: data curation. Stefano Lorito: supervision, methodology, writing
- review and editing.

. **Competing interests.** The contact author has declared that none of the authors has any competing interests.

. **Acknowledgements.** The authors would like to thank Christopher Moore, for the insightful discussions regarding DART stations and the properties of tsunami detection algorithms, and the two anonymous reviewers for their insightful comments and suggestions. The work has been carried out as part of the collaboration "Accordo di collaborazione tra DIFA e INGV Roma con Rep. 51/2020 Prot. 803 del 08/05/2020".

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
