# Peer review of "Tsunami detection methods for Ocean-Bottom Pressure Gauges"

_Natural Hazards and Earth System Sciences, 2024_

## Author Response (AR1)

We would like to thank the two anonymous reviewers for the interesting and insightful comments about our work. In the following, we present a point-by-point response to the comments.

**Response to Reviewer 1**:

**1. In the Introduction section, it is unnecessary to elaborate extensively on tide gauges because they do not have much significance in tsunami detection. Instead, the focus should be on offshore observations. For instance, I suggest categorizing OBPG systems; those use cable for signal transmission, while others use sonar (DARTs). Additionally, the author could discuss GPS buoys.**

- Although we do think that tide gauges can be important for detection, e.g. in the case of late arrivals or to alert coastal areas further away, they do not have any role in the present work, thus we will modify the text, following your suggestions. The discussion on tide gauges has been greatly reduced, while introductory information about the different types of OBPGs have been expanded. We also added a brief mention to GPS buoys.

**2. In the application of real-time tsunami detection, the computational speed of algorithms is also crucial. I suggest the authors discuss the computational efficiency of these methods and whether they can rapidly produce results.**

- All the algorithms are already fast enough to be used in real time, i.e. the computational costs for each time step is shorter than the sampling time of the instruments. Nonetheless, we understand the need for a more detailed analysis, presented in section 3.2, where the computation time of the technique have been computed for one of the examples and used to compare the techniques.

**3. When DART records sea surface elevation data, significant deviations can sometimes occur. I would like the authors to discuss how to address such situations.**

- We do recognize that the direct conversion of pressure data to sea level data as was done in the paper is valid only within the boundary of the shallow water approximation and it does not hold in many situations, such as coupled air sea waves. In the introductory part of section 3, we acknowledge this possibility, but it is not taken into account in the analysis. The reason is that the detection algorithms are applied to raw pressure data, thus its exact relation with sea level is not of concern. Nonetheless, it is very important to consider this if any of the detection methods is integrated into an early warning system.

**4. In Line 105, I disagree with this viewpoint; even within the same bay, due to tsunami resonance, long-period signals can exhibit significant differences at different locations.**

- This is in fact not a general statement, but it is experimentally valid for the EOF detiding as claimed by Tolkova (2009, 2010). We specified more clearly the origin and scope of validity of the statement in the section where EOF is introduced. Furthermore, it looks like the technique has worse performance on some DART stations (located in New Zealand)

that were added to the background analysis section as part of the suggestions from Reviewer 2.

**5. Regarding FTF-based decomposition, what causes significant false signals to appear before the arrival of a tsunami (Figure 2c)? How can this be avoided?**

- The appearances of this oscillations before the real disturbance in the decomposition is related to the loss of causality intrinsic in signal decomposition techniques. This is not exclusive to IMF decompositions, since it can be observed also in the case of classical Fourier trigonometric series and it can be observed in Fig. 18 and 19 of the work by Tolkova (2009). In the specific case, the presence of a "spike" at the end of the signal is the specific cause of these oscillations. By summing the components within a not too narrow frequency band, their effect is cancelled out, as shown in Fig. 3c. A comment on this has been added in the relevant section

**6. In Line 274, how is the post-processed waveform obtained? Please provide a detailed explanation.**

- The post-processed waveforms are taken directly from the material provided by Davies (2019). To remove the tidal component a LOWESS filter was applied, while seismic oscillations were removed by truncation. A brief explanation of how they are obtained has been added to the paper.

**Other minor comments:**
**1. Figure 1: Please add the unit to y-axis**

- Being a basis set, the computations do not change if the vectors are multiplied by an arbitrary constant. For this reason, a unit of measurement is not strictly necessary. The same holds for the plot ranges. Nonetheless, we see that in the present state, the text and this figure may be disorienting. So, the suggestions about the plot has been implemented and an explanation about the arbitrariness of amplitude of these vectors will be added to the text.

**2. Line 190: Do you refer to , "exclude detiding procedures"? Any typo?**

- In that case, we refer to techniques which remove tides with a model based approach, which in this work are EOF and TDA. In cases where the model fails to account for some tidal components correctly, these will have an effect on the detection curves. The text has been reworded.

**3. Line 322: ,"much better then" -> "much better than"**

- The text has been corrected.

**Response to Reviewer 2**:

**1. Introduction should state clearly the motivation and scientific questions for why the work is being done.**

- The motivation behind the work is to test Tsunami detection algorithms. More reasoning on this aspect has been added to the end of Introductory section, in particular in terms of the characteristics and properties of the techniques that we investigate in later sections. The space dedicated to discussion of coastal sea level stations has been also reduced and more details about the various types of pressure gauges have been added.

**2. The work will benefit greatly by comparing computational costs of each of the four algorithms. A robust method that takes up too much time defeats the purpose of real-time detection. The authors say the algorithms are efficient, but it should also be shown. This addition would make the paper more impactful.**

- Each technique is fast enough to be used in real time, i.e. the computational costs for each time step is shorter than the sampling time of the instruments. Nonetheless, we understand the need for a more detailed analysis, presented in paragraph 3.2.

**3. Background noise should be done for the same time periods rather than at differing times. This leaves the DART stations open to misrepresented background values that may be influenced by meteorological and oceanographic phenomena. All 5 DARTs studied here have overlapping data at the same times with multiple tsunami detections in those times. Without accounting for these features, it makes the research less impactful, as it remains unclear if differing conditions favor different detection algorithms.**
    **1 The manuscript should add the New Zealand DARTs, as they provide a good dataset for the Southern Pacific.**

- The major difficulty for the analysis of overlapping time series in different DART is finding portions of data as close to "ideal" as possible to reduce the influence of other factors. In particular, as specified in the paper, we want weeks long time series with no seismic shaking, no tsunami signal, no discontinuity, no holes in the data and enough preceding data to compute accurate tidal coefficients for TDA. This last point is the most critical. Nonetheless, we were able to find a 20-day period window (08/04/2021 – 27/04/2021) where 16 DARTs were recording in "background" conditions simultaneously. Their analysis has been added to the background analysis section. We note that these 16 DARTs also include instruments from New Zealand DARTs, as suggested.

**Minor comments**

**1. Line 46-47: These are not review papers but proper studies. Refer to them as such.**

- The text has been reworded accordingly

**2. Line 49: The proper citation for the DART network is Titov, V.V., F.I. González, E.N. Bernard, M.C. Eble, H.O. Mofjeld, J.C. Newman, and A.J. Venturato (2005): Real-time tsunami forecasting: Challenges and solutions. Hazards, 35(1), Special Issue, U.S. National Tsunami Hazard Mitigation Program, 41–58.**

- The paper was already cited elsewhere in this work, and we now corrected the citation in line 49 as well.

**3. Cite DART data as National Oceanic and Atmospheric Administration (2005): Deep-Ocean Assessment and Reporting of Tsunamis (DART(R)). NOAA National Centers for Environmental Information. doi:10.7289/V5F18WNS [access date].**

- The citation has been added where DART stations are introduced.

**4. Line 72: Do not use DART for more than one acronym. It is confusing, and it muddies the narrative. Instead, call it MOF for Mofjeld (1997).**

- The text and figures have been modified accordingly.

**5. Line 86: Be consistent with use of "on board" or "on-board." Most instances of the use of "on board" are incorrect.**

- The text has been corrected.

**6. Line 106: State the length of a lunar day in other time units (e.g., hours, minutes, or seconds).**

- The lunar day is 24h 50.4min, as reported by Tolkova (2009, 2010). The information has been added to the text in the relevant section.

**7. Figure 1: Missing y-axis units. Also, make y-axis the same limit for all subplots.**

- Being a basis set, the computations do not change if the vectors are multiplied by an arbitrary constant. For this reason, a unit of measurement is not strictly necessary. The same holds for the plot ranges. Nonetheless, we see that in its former state, the text and this figure may be disorienting. So, the suggestions about the plot have been implemented and an explanation about the arbitrariness of amplitude of these vectors has been added to the text.

**8. Line 144: Fix the units in the brackets.**

- The units have been fixed.

**9. Line 175: Why do you choose this range of periods?**

-   We choose this range in order to keep in the oscillation in the tsunami frequency band. Nonetheless, we should also note that the technique is quite robust in changing these parameters, in particular the larger period. This explanation has been added to the text.

**10. Line 196: Colloquial speak — replace with a different, appropriate modifier.**

-   The text has been reworded

**11. Line 200: Replace "taller" with larger.**

-   Done.

**12. What does it mean to be a "long" deployment?**

-   In this context, a "long" deployment is one that is long enough to compute a tidal model accurately for the TDA technique, so at least several months. This explanation has been added to the text.

**13. Figure 7: This figure needs a long caption to explain all of what is going on in it. What is the x-axis? Tidal coefficients? If so, mention it! Do not assume everyone knows what they are. Also, choose a better colorblind friendly color scheme. I am lost as to what is being plotted here because they all look the same.**

-   The plot shows absolute average value for each time series in the dataset, thus each tick on the x axis represents one of the signals. A detailed description of the plot has been added to the caption. The color palette has been updated in all figures where it was needed.

**14. Same as Figure 7 for Figure 8. It needs to be clearer that the x-axis is time not tidal coefficients. DO NOT leave it in the text!**

-   The answer to comment 13 holds here as well.

**15. Line 325: DARTs should be referred to as DART stations rather just DART buoys. DART buoy refers to the buoy, and it neglects the bottom pressure recorder. DART station is all inclusive of the whole system.**

-   Every occurrence has been corrected.

**16. Line 335: "Remains the best performing ..." is a less awkward turn of phrase.**

-   The correction has been applied.

**17. Line 361: Authors should consider posting FIR filter coefficients and EOF scripts as they were calculated using MATLAB with examples to ensure correct application. I was able to check everything else; however, good scientific coding practices dictate that even if native functions are used that they be viewable to ensure correct application of them.**

- The codes for both of them are very simple and only use basic MATLAB functions. All the information needed for the implementation are present in the paper. Nonetheless, the codes are added as Supplementary Material.